# Antioxidant Carboxymethyl Chitosan Carbon Dots with Calcium Doping Achieve Ultra-Low Calcium Concentration for Iron-Induced Osteoporosis Treatment by Effectively Enhancing Calcium Bioavailability in Zebrafish

**DOI:** 10.3390/antiox12030583

**Published:** 2023-02-26

**Authors:** Lidong Yu, Xueting Li, Mingyue He, Qingchen Wang, Ce Chen, Fangshun Li, Bingsheng Li, Li Li

**Affiliations:** 1School of Life Science and Technology, Harbin Institute of Technology, Harbin 150080, China; 2School of Physics, Harbin Institute of Technology, Harbin 150080, China; 3Key Laboratory of UV Light Emitting Materials and Technology of Ministry of Education, Northeast Normal University, Changchun 130024, China

**Keywords:** iron overload, osteoporosis, carbon dots, antioxidant, calcium bioavailability

## Abstract

Iron overloads osteoporosis mainly occurs to postmenopausal women and people requiring repeated blood transfusions. Iron overload increases the activity of osteoclasts and decreases the activity of osteoblasts, leading to the occurrence of osteoporosis. Conventional treatment options include calcium supplements and iron chelators. However, simple calcium supplementation is not effective, and it does not have a good therapeutic effect. Oxidative stress is one of the triggers for osteoporosis. Therefore, the study focuses on the antioxidant aspect of osteoporosis treatment. The present work revealed that antioxidant carboxymethyl chitosan-based carbon dots (AOCDs) can effectively treat iron overload osteoporosis. More interestingly, the functional modification of AOCDs by doping calcium gluconate (AOCDs:Ca) is superior to the use of any single component. AOCDs:Ca have the dual function of antioxidant and calcium supplement. AOCDs:Ca effectively improve the bioavailability of calcium and achieve ultra-low concentration calcium supplement for the treatment of iron-induced osteoporosis in zebrafish.

## 1. Introduction

Osteoporosis (OP) is a disease characterized by reduced bone mass, damaged bone microstructure, increased bone fragility, and easy to fracture bones [1,2,3]. A bone is a metabolically active tissue. Bone homeostasis is regulated by osteoblast-mediated bone formation and osteoclast-promoted bone resorption. The disruption of bone homeostasis performs a fundamental role in the pathogenesis of OP [3]. At present, post-menopausal women are the main population of OP. Some studies have suggested that post-menopausal women OP is not only related to decreased estrogen, but also may be the result of elevated iron levels. An important way to maintain the balance of iron is menstrual bleeding to women. The ferritin increases about 2–3 times from the perimenopausal period of 45 years old to the post-menopausal period of 60 years old [4]. Clinical treatments include bis-phosphonates, estrogens and related compounds, vitamin D, and calcitonin [5]. In particular, the combination of calcium supplements with vitamin D is considered a routine strategy for the treatment of OP [6,7]. However, taking calcium supplements to prevent and treat OP has become a controversial topic. Calcium supplementation may fail to compensate for renal calcium loss, and increased calcium load in circulation could lead to extra skeletal deposition, including in the coronary arteries [8,9].

Antioxidant drugs for osteoporosis have become a new research hotspot. Studies have found that most OP patients show high levels of oxidative stress. Excessive ROS upset the balance between bone formation by osteoblasts and resorption by osteoclasts, leading to bone mass loss and bone quality degradation. Therefore, an oxidative stress injury is considered a major pathogenesis of osteoporosis [10]. Iron ions generate ROS through the Fenton reaction, which enhances the activity of osteoclasts and decreases the activity of osteoblasts. Oxidative stress induced by iron overload can lead to impaired bone formation and mineralization, and damage the microstructure of bone tissue and promote disease progression [11,12,13,14]. The increase in iron level in post-menopausal women may be one of the causes of OP. Estrogen inhibits iron-regulator synthesis, maintains iron transporter protein integrity and enhances iron uptake by duodenal enterocytes and iron release from iron storage macrophages and hepatocytes. Therefore, postmenopausal women exhibit iron accumulation outside of estrogen deficiency. Iron accumulation promotes bone resorption and bone loss through oxidative stress and inflammatory responses [13,15,16,17,18]. There are three main types of antioxidant defense systems that eliminate free radicals and peroxides, including antioxidants (Glutathione, Melatonin, Vitamin C, and Vitamin E, Protein Antioxidants: Ferritin and Ceruloplasmin), anti-oxidase (Superoxide Oxidoreductase, Catalase, and Glutathione Peroxidase), and repair enzyme (DNA Repair: Glycosylase, AP-Endonuclease, and DNA Polymerase, Lipid Peroxide Metabolism: Phospholipase A2, and Acyltransferase) [19,20]. Therefore, improving antioxidant capacity and removing excessive ROS are also effective methods of the treatment of OP. Many carbon dots (CDs) with antioxidant properties have been reported to relieve oxidative stress in zebrafish [21,22,23]. At the same time, there are also many CDs with iron chelating function [24,25,26]. This means CDs could be a potential drug for the treatment of OP induced by iron overload.

Zebrafish is an advantageous model organism for OP research and drug screening. The first cartilage structure of the jaw is formed at 2-day post fertilization (dpf) in zebrafish. The cleithrum, endopterygoid, sphenoidalia, and other structures of the head can be observed by alizarin red staining at 6 dpf. In this study, 0.2 mM ferrous ammonium sulfate (FAS) was selected to establish iron overload OP model in zebrafish larvae. Iron chelated EWCDs [24] and antioxidant carboxymethyl chitosan-based carbon dots (AOCDs) were used to treat iron overload OP. It was found that iron overload OP was not alleviated by iron chelated EWCDs. Antioxidant AOCDs had good therapeutic effect. In addition, calcium-doped carboxymethyl chitosan carbon dots (AOCDs:Ca) had a better therapeutic effect than AOCDs and calcium gluconate salt. AOCDs:Ca superimposes the two functions of antioxidant and calcium supplementation, which greatly improves the bioavailability of calcium. AOCDs:Ca is a potential drug for the treatment of iron overload OP.

## 2. Materials and Methods

### 2.1. Zebrafish Rearing and Mating

AB zebrafishes were provided by the Aquatic Institute of Heilongjiang. The zebrafish was raised in the (AAE-022-AA-A) circulating water system. The water temperature was 28 °C, and the photoperiod was 14 L:10 D. The zebrafish were fed twice a day with brine shrimp. Adult zebrafish mated in a 2:1 ratio of male to female. The time after fertilization was denoted as hour post-fertilization (hpf). 

### 2.2. Sample Collecting

Zebrafish embryos were randomly placed in 6-well culture plates (30 embryos in 4 mL solution per well). Deionized water was added to the control group. The model group was added 0.2 mM ferrous ammonium sulfate (FAS, Aladdin). The solution was changed every 24 h. Zebrafish embryos in FAS were reared at 28 °C for 48 h. Then, embryos were rinsed with deionized water. Deionized water was added to the control group. To the model group, we added 1 mg mL^−1^ carbon dots solution or 0.6287 μg mL^−1^ calcium gluconate at 48 hpf without a water change. Zebrafish larvae were raised to 6 dpf. A total of 30 larvae were used for determination of ROS and ALP content. A total of 30 larvae were used for RNA extraction. Several larvae were taken and fixed overnight with 4% PFA at 4 °C for alizarin red staining. Alizarin red was purchased from Tianjin Zhiyuan Chemical Reagent Co., LTD. (Tianjin, China)

### 2.3. Quantitative Real-Time PCR

Total RNA was isolated using TRIzols reagent (Takara, Dalian, China). cDNA was obtained by PrimeScriptTM RT Reagent Kit (Takara, Dalian, China). Real-time PCR was performed with 7500 Real-Time PCR SYSTEM (Applied Biosystems, Foster City, CA, USA) by using SYBR Premix Ex Taq II (Takara, Dalian, China) according to the recommended instructions. The primer sequences are shown in Appendix A. Three parallel experiments were set up for each sample. Ef1α was used as a housekeeping gene. The experimental data were processed by the 2^−ΔΔCT^ method. 

### 2.4. Alizarin Red Staining

Zebrafish larvae were fixed with 4% PFA. Then, zebrafish larvae were washed twice with 1× PBS. Zebrafish larvae were depigmented with 1.5% H_2_O_2_/0.25% KOH solution. Zebrafish larvae were washed with 25% glycerol/0.1% KOH solution. A total of 0.05 % alizarin red solution was prepared and stained for 30 min in darkness. The 50% glycerol/0.1% KOH solution was prepared to wash larvae, until the solution had no obvious color. A total of 50% glycerol was added, and a Macro zoom fluorescence microscope was used to photograph (Axio zoom. V16, ZISS). Image analysis software (Image J 1.52A) was used to calculate the staining area.

### 2.5. The Synthesis of AOCDs and AOCDs:Ca

A total of 0.1 g Carboxymethyl chitosan and 2 g acrylamide, or 0.1 g Carboxymethyl chitosan, 2 g acrylamide, and 0.0448 g calcium gluconate salt (Aladdin, Shanghai, China) was dissolved in 20 mL deionized water, and microwaved 700 W for 10 min. The products were dissolved in 30 mL deionized water and centrifuged at 12,000 rpm for 20 min. The supernatant was filtered with 0.22 µM filter membrane and dialyzed with 1000 Da for 48 h. The deionized water was changed every 12 h. The powder of AOCDs and AOCDs:Ca was obtained by freeze drying. 

### 2.6. The Characterization of AOCDs and AOCDs:Ca

UV-vis absorption spectra were performed using a SHIMADZU UVmini-1240 UV-VIS spectrophotometer. PL spectra and lifetime were measured by state/transient fluorescence spectrometer (FLS1000, Edinburgh). FT-IR spectra were taken on a Nicolet IS50 (Thermo Fisher, Waltham, MA, USA) FT-IR spectrophotometer. The TEM and HR-TEM images were recorded using FEI, TECNAI TF20 field emission electron microscope. X-ray photoelectron spectroscopy (XPS) experiments were performed using electronic spectrometer (ESCALAB XI+, Thermo Fisher). The results (binding energies) were calibrated using the C 1s peak for C-C at 284.8 eV, and fitted using the software Thermo Advantage v5.967 (Thermo Fisher Scientific Inc., USA). The determination of calcium content was performed by Inductively coupled plasma spectrometer (iCAP 7400, Thermo Fisher).

### 2.7. The Biosafety Assessment of AOCDs and AOCDs:Ca

Zebrafish embryos were exposed to 0 mg/mL, 0.1 mg mL^−1^, 0.2 mg mL^−1^, 0.4 mg mL^−1^, 0.8 mg mL^−1^ AOCDs, or AOCDs:Ca solutions at 3–4 hpf. A total of 30 zebrafish embryos were prepared in each group. Each group had three repetitions. Dead embryos were removed, and the solution was changed every 24 h. We measured the frequency of autonomic movement at 24 hpf for 1 min, heart rate at 48 hpf for 15 s, the hatching rate of zebrafish embryos at 54 hpf. The survival rate and malformation rate of zebrafish embryos was evaluated at 96 hpf. Among them, the frequency of autonomic movement was used to characterize the neurotoxicity of CDs. The heart rate was used to characterize the toxicity of blood circulation system of CDs. The hatching rate was used to characterize the impact of CDs on development. The malformation rate and death rate directly reflected the biological safety of CDs. 

### 2.8. Determination of ROS

Live Zebrafish were added with 0.1 mM DCFH-DA, kept in the dark for 1 h, and washed with DW. After zebrafish embryos were anesthetized with 0.05% Tricaine for 5 min, the zebrafish were observed and photographed by fluorescence microscope.

### 2.9. Determination of ALP

A total of 30 zebrafish embryos were collected and homogenized with 0.9% NaCl. Centrifugation was performed at 1000 g/min at 4 °C for 10 min. The total protein assay kit (BCA method) was purchased by beyotime (Shanghai, China). The content of ALP was measured using Kit (Jian cheng, Nanjing, China).

### 2.10. Determination of Ca in AOCD:Ca

AOCD:Ca were taken and treated with chloroazide (HNO_3/_HCl = 1/3, *v*/*v*) at 70 °C for 24 h. The content of AOCD:Ca with Ca^2+^ in the samples was determined by ICP-OES.

### 2.11. Data Analysis

The stained area was calculated by AI. All data were expressed as mean ± SEM from at least three independent experiments. Group differences were performed using *t*-test. * *p* < 0.05, ** *p* < 0.01, *** *p* < 0.001 and **** *p* < 0.0001 were considered statistically significant by GraphPad Prism 6 and Origin 8 software.

## 3. Results

### 3.1. Establishment of Iron Overload Osteoporosis Model

First, the OP model of zebrafish was successfully established by using 0.2 mM of FAS. The use of FAS has almost no toxic effects and does not affect the cartilage development of zebrafish embryos. Images of juvenile zebrafish after alizarin red head skeletal staining showed that bone loss and a significant decrease in bone mineral density after FAS treatment. Importantly, after numerical analysis of the images, it was found that the staining area was reduced compared to the control group (Figure 1A,B). These results exhibit osteoporotic symptoms after the appearance of FAS treatment. To reconfirm the OP model, the expression of osteoblast- and osteoclast-related genes was examined at the molecular level. Compared with the control group, the expression of *runx2b*, a marker gene of osteoblasts, was significantly decreased after FAS treatment, while the expression of *bmp2b* was not significantly changed (Figure 1C,D). On the other hand, since mature osteoblast tissues contain large amounts of alkaline phosphatase (ALP), ALP content can be used as a marker of osteoblast tissue maturation. Compared with the control group, ALP content was significantly decreased after FAS treatment (Figure 1E). These results implied that FAS inhibited the expression of *runx2b* and decreased the content of osteogenic tissue. In addition, Cathepsin K (*ctsk*) was abundant in osteoblasts and performed an important role in bone resorption by osteoclasts. Meanwhile, anti-tartrate phosphatase type 5 (*acp5b*) is a commonly used marker of osteoclasts and bone resorption. The mRNA expression levels of *ctsk* and *acp5b* were significantly increased after FAS treatment compared to controls, indicating active osteoclasts (Figure 1F,G). Ultimately, these results demonstrate that FAS can be used to establish a zebrafish OP model.

### 3.2. Preparation and Characterization of the AOCDs and AOCDs:Ca

Using carboxymethyl chitosan (CC) and acrylamide (AM) as raw materials or doped calcium gluconate as raw materials, AOCDs and AOCDs:Ca were obtained by hydrothermal treatment. The solution appeared yellow under sunlight and had blue fluorescence emission at UV-365 nm (Figure 2A). Transmission electron microscopy (TEM) showed that AOCDs and AOCDs:Ca were uniformly divided in deionized water with average particle sizes of 4.91 nm and 4.99 nm. The diffraction peak was clearly observed under high resolution TEM (HRTEM) (Figure 2B,C). The diffraction peak spacing was 0.20 nm and 0.22 nm, respectively, corresponding to the 100 and 101 faces of graphite. These results indicated the successful preparation of graphene quantum dots. Based on the fluorescence luminescence properties of CDs, we evaluated the optical properties of AOCDs and AOCDs:Ca by PL emission. It was found that AOCDs and AOCDs:Ca had excitation wavelength-dependent fluorescence emission properties. Optimal excitation and emission were 370 nm and 450 nm, respectively (Figure 2D). At the same time, AOCDs and AOCDs:Ca were observed to redshift about 80 nm and 73 nm, respectively, with the increase in excitation light by normalization (Figure 2E). The UV-vis spectrum showed a broad absorption range, indicating that AOCDs and AOCDs:Ca had aromatic-like structure. The absorption shoulder at 257 nm was caused by the π-π* transition of C=O/C=C (Figure 2F). The absorption of AOCDs:Ca was significantly weaker than that of AOCDs, indicating successful Ca^2+^ doping. The fluorescence lifetime of these two CDs was 5.59 ns and 5.15 ns, respectively (Figure 2G). 

In order to further study AOCDs and AOCDs:Ca functional groups on the surface, we analyzed the AOCDs and AOCDs:Ca chemical structure by X-ray photoelectron spectroscopy (XPS) and Fourier transform infrared (FT-IR) spectra. The full XPS spectra of AOCDs showed three peaks of 284.57, 398.65, and 532.85 eV, respectively, indicating that AOCDs was composed of C, O, and N elements with atomic ratios of 65.11%, 18.19%, and 16.69%, respectively (Figure 3A). The high-resolution XPS spectra of the C 1s band were divided into three peaks at 284.80, 285.75, and 287.87 eV, corresponding to C-C/C=C, C-N/C-O, and C=O, respectively (Figure 3B) [27,28,29]. N 1s band had one peak located at 399.57 eV, which belonged to Pyridine N (Figure 3C) [30]. The O 1s band had two peaks at 531.08 and 531.91 eV, corresponding to C=O and O-C/O-H, respectively (Figure 3D) [31,32]. The total XPS spectra of AOCDs:Ca showed 284.57, 398.65, 532.85, and 346.78 Ca2p eV peaks, respectively, indicating that AOCDs:Ca was composed of C, O, N, and Ca elements, with atomic ratios of 49.09%, 33.23%, 17.07%, and 0.61%, respectively (Figure 3E). The high-resolution XPS energy spectra of C 1s band was divided into three peaks at 284.80, 285.82, and 287.80 eV, corresponding to C-C/C=C, C-N/C-O, and C=O, respectively (Figure 3F) [33,34]. N 1s band had one peak located at 399.55 eV, which belonged to Pyridine N (Figure 3G) [35]. The O 1s band had two peaks at 531.17 and 532.07 eV, corresponding to C=O and C-O, respectively (Figure 3H) [36,37]. The Ca 2p band had two peaks located at 347.03 and 350.68 eV (Figure 3I). Together, these four high-resolution spectra showed that Ca atoms had been successfully embedded in AOCDs:Ca and existed as ions. This provided the basis for the efficient release of Ca^2+^. In addition, FT-IR spectra showed that the stretching vibration of O-H/N-H from CC of 3291 cm^−1^, the stretching vibration of carboxyl (1650 and 1438 cm^−1^), and amide carbonyl from AM (C=O, 1650 cm^−1^) in AOCDs. The mixed in-plane bending vibration of amides C-H and N-H was located at 1436 cm^−1^. Therefore, AOCDs and AOCDs:Ca are obviously hydrophilic. Based on the above characterization results, Ca^2+^ doped success, AOCDs:Ca have aromatic structure similar to AOCDs. It also has the optical properties of the origin of the defect of pyridine nitrogen atom (Figure 3J). 

### 3.3. Antioxidant Properties of AOCDs and AOCDs:Ca

The biosafety of AOCDs and AOCDs:Ca was determined. Zebrafish embryos were treated with different concentrations of AOCDs and AOCDs:Ca. The frequency of autonomic movement, heart rate, hatching rate, and survival rate had no significant difference. The malformation rate was lower (Appendix A). In order to ensure the safety of CDs, 0.1 mg/mL was selected as the treatment concentration. We detected whether ferroptosis occurred in OP caused by iron overload, and whether CDs alleviated FAS-induced oxidative stress, Glutathione peroxidase (*gpx4a*), and ferritin (*fth1*) inhibit ferroptosis. In zebrafish, they correspond to *gpx4a* and *fth1a*. Compared with control group, the expression level of *gpx4a* and *fth1* were significantly increased in FAS group. This result indicated that ferroptosis did not occurred in iron overload OP. Compared with FAS group, the expression level of *gpx4a* did not change significantly in EWCDs group but increased significantly in the AOCDs and AOCDs:Ca groups. Upregulation of *gpx4a* expression implied that AOCDs and AOCDs:Ca had antioxidant properties. Compared with FAS group, the expression of *fth1a* did not change significantly in AOCDs and AOCDs:Ca groups. The down-regulation of *fth1a* expression by EWCDs may be due to its iron chelation ability (Figure 4A,B). Further detection of ROS showed that the levels of ROS was significantly increased after FAS treatment, which triggered oxidative stress in zebrafish. EWCDs had no effect on ROS content compared with FAS group. After treatment with AOCDs and AOCDs:Ca, the content of ROS decreased significantly (Figure 4C,D). These results indicated that EWCDs had no antioxidant properties, while both AOCDs and AOCDs:Ca had antioxidant properties, AOCDs had better antioxidant performance.

### 3.4. AOCDs:Ca Improve the Effect of Iron Overload Osteoporosis Treatment

We have confirmed that EWCDs alleviates iron-overloaded nonalcoholic fatty liver disease through Fe^3+^ chelation [24]. Therefore, this study attempted to use EWCDs, AOCDs, and AOCDs:Ca in the treatment of iron overload OP. Compared with the control group, the alizarin red staining found that the dyeing area of FAS group was reduced. Compared with the FAS group, the EWCDs group had no obvious change. AOCDs and AOCDs:Ca groups increased (Figure 5A). The stained area represents bone mass. Numerical analysis of the staining results showed that there was no significant change after EWCDs treatment, but significant increase in bone mass after AOCDs and AOCDs:Ca treatment (Figure 5B). These results indicated that EWCDs had no effect, whereas both AOCDs and AOCDs:Ca had effect on iron overloaded OP. 

Treatment efficacy was assessed at the molecular level. Compared with FAS group, the content of ALP increased after EWCDs and AOCDs treatment but did not change significantly after the AOCDs:Ca treatment (Figure 5C). *Runx2b* was not affected after treatment with the three CDs (Figure 5D). The expression of *bmp2b* increased after AOCDs treatment. EWCDs and AOCDs:Ca had no effect on *bmp2b* (Figure 5E). At the same time, we found that the expression of *ctsk* was increased by EWCDs and AOCDs, compared with the FAS group. The AOCDs:Ca significantly decreased the expression of *ctsk* (Figure 5F). The level of *acp5b* was decreased by the three CDs (Figure 5G). It is well known that the metabolic balance of bone is maintained by osteoblasts and osteoclasts [11]. Although EWCDs promoted the production of ALP, it also up-regulated the expression of osteoclast gene. This may be the reason why EWCDs does not treat iron overload OP. The AOCDs up-regulated the expression of *bmp2b*, down-regulated the expression of *acp5b*, and promoted the production of ALP. The AOCDs promoted the balance of bone metabolism toward bone remodeling to achieve the treatment of iron overload OP. Although AOCDs:Ca only inhibited osteoclasts gene (*ctsk*, *acp5b*) expression, alizarin red staining was deepened after treated with AOCDs:Ca. It indicated the content of calcium increased. Due to the doping of calcium, the AOCDs:Ca achieved calcium supplementation. The AOCDs and AOCDs:Ca significantly reduced the ROS content and effectively alleviated iron overload OP. Although calcium doping attenuated the antioxidant effect, it made up for the deficiency of antioxidant through calcium delivery. Therefore, the AOCDs:Ca had the best therapeutic effect on iron overload OP. 

### 3.5. AOCDs:Ca Improve Calcium Bioavailability

Next, we compared the therapeutic effects of calcium supplementation and AOCDs:Ca. The calcium content of AOCDs:Ca was determined by ICP. It was found 1 mg mL^−1^ AOCDs:Ca contained 0.6287 µg mL^−1^ calcium (Figure 6A). It was consistent with the calcium content measured by XPS (Figure 3E). We used the same concentration of 0.06287 µg mL^−1^ Ca^2+^ to treat iron overload OP. Low calcium supplementation alone did not improve osteoporosis symptoms compared with FAS group. However, AOCDs:Ca with low concentrations of calcium were recovery (Figure 6B,C). This indicated that AOCDs:Ca not only alleviated iron overload OP through anti-oxidation, but also improved the bioavailability of calcium. At the mRNA level, calcium supplementation treatment had no effect on the recovery of *bmp2b* and *acp5b*, although there was some recovery of *runx2b* and *ctsk* (Figure 6D–G). In contrast, AOCDs:Ca inhibited osteoclast gene expression on the one hand and increased calcium content on the other. Together, it promoted the balance of bone metabolism towards bone remodeling, and thus the treatment of iron overload osteoporosis. Thereafter, it had better therapeutic effect than calcium supplementation alone and AOCDs.

## 4. Discussion 

FAS was selected to establish the iron overload OP model in zebrafish. This is because FAS can better cause reduced ossification without affecting the development of zebrafish, thus creating a model of osteoporosis (Appendix A). When iron overload is mentioned, the first thing that comes to mind is treatment with iron chelators. However, chelators are often accompanied by side effects [38,39,40]. Therefore, it is an important research topic to find a safer and more effective iron chelator.

The small size of CDs makes It easy to enter zebrafish, it can be metabolized, and it has high biosafety [41]. EWCDs with excellent iron chelation function are selected. However, EWCDs have no therapeutic effect on OP caused by iron overload. The levels of ROS increased in iron overload OP, indicating that iron overload caused oxidative stress. EWCDs did not reduce the ROS content. Interestingly, when we turn our attention to AOCDs, we find that AOCDs effectively clear iron-induced ROS. It blocks the oxidative stress and lipid peroxidation caused by iron overload and alleviate the OP caused by oxidative damage [42]. It indicated that oxidative stress was the main reason of iron overload OP. Antioxidant performed an important role in treatment of iron overload OP. In addition, AOCDs improve ALP activity and stimulate osteoblast differentiation by upregulating related mRNA expression. This suggests that AOCDs alleviate iron overload OP by promoting bone formation. OP is also characterized by overactivity of osteoclasts. Inhibition of bone resorption by hyperactive osteoclasts may be a potential therapeutic strategy, but there are currently no effective drugs in clinical practice [43]. Selective regulation of Ca^2+^ can affect the balance of bone metabolism and regulate the proliferation, differentiation and function of osteoblasts/osteoclasts [44]. We determine that AOCDs serve as nanocarriers for Ca^2+^. Although Ca doping leads to a decrease in antioxidant capacity, the staining deepens after AOCDs:Ca treatment, indicating an increase in intraosseous calcium content. It may be the main reason for its better effect than AOCDs. To confirm the biological function of calcium, we use AOCDs:Ca and calcium gluconate with the same calcium content for the treatment of iron overload OP, respectively. Calcium supplementation alone is found to be ineffective, suggesting that the superior therapeutic effect of AOCDs:Ca on iron overload osteoporosis results from increased calcium bioavailability. Therefore, the antioxidant properties of AOCDs:Ca and the promotion of low-dose calcium delivery and absorption make it a potential drug for OP treatment.

Studies have shown that drug-loaded nanomaterials or CDs can increase the concentration of drugs through sustained release. However, whether AOCDs:Ca can sustain the release of Ca^2+^ to enrich Ca^2+^ at the site of OP, and thus improve the bioavailability, remains to be further investigated. Furthermore, it is not known which antioxidant signaling pathway or antioxidant defense system is activated in the treatment of iron overload OP. In addition, rheumatoid arthritis (RA) patients often accompany the occurrence of OP [45]. Most importantly, AOCDs themselves have the function of treating OP. Using AOCDs as an anti-RA drugs vector, a new drug can be developed to treat both RA and OP. This could be a future direction for AOCDs.

## 5. Conclusions

In summary, we successfully constructed a novel functionalized carbon dots as a calcium supplement to achieve the treatment of osteoporosis. Here, carboxymethyl chitosan-based carbon dots (AOCDs) were found to alleviate iron overload osteoporosis as an antioxidant. Meanwhile, with the introduction of calcium ions (AOCDs:Ca), the dual effect of antioxidant and calcium supplementation was achieved, effectively treating iron-induced osteoporosis in zebrafish. More importantly, the therapeutic effect of AOCDs:Ca was superior to that of calcium gluconate of the same quality. AOCDs:Ca effectively improved the bioavailability of calcium and achieved an ultra-low concentration of calcium supplementation. Thus, this work provides new insights to optimize high quality antioxidants and to develop an efficient calcium supplementation. With further optimization, this strategy shows great potential for application to the alleviation of various types of calcium deficiencies.

## Figures and Tables

**Figure 1 antioxidants-12-00583-f001:**
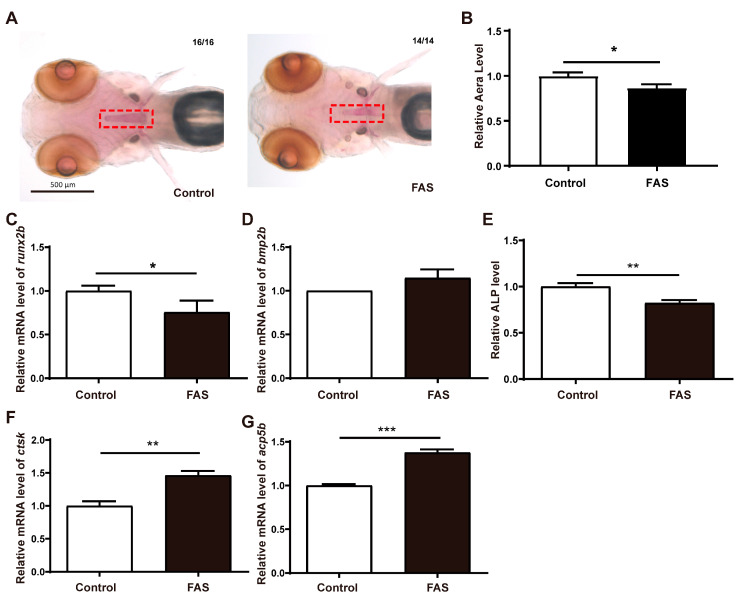
Establishment of iron overload osteoporosis model. (**A**). Alizarin red staining, The red rectangle is the location of osteoporosis. (**B**). Numerical analysis of staining area (unit: pixel), Effect of FAS on genes osteoblast-regulated genes, (**C**). *runx2b*, (**D**). *bmp2b*, (**E**). The content of ALP, Effect of FAS on osteoclast-regulated genes, (**F**). *ctsk*, (**G**). *acp5b*, (*n* = 3, *t*-test, * *p* < 0.05, ** *p* < 0.01, *** *p* < 0.001).

**Figure 2 antioxidants-12-00583-f002:**
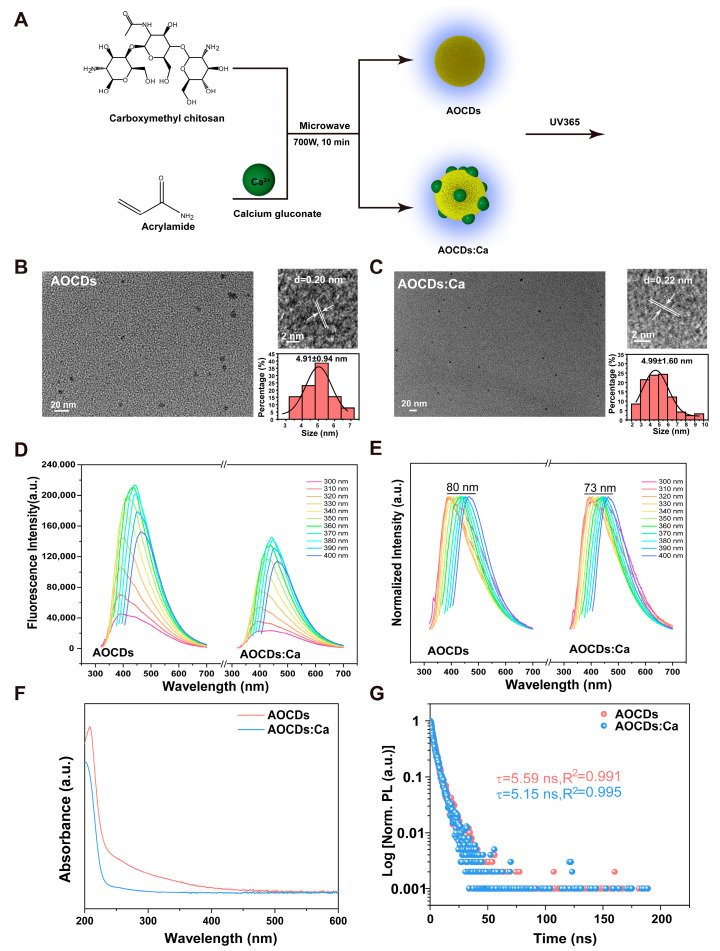
Particle size and luminescence characteristics of CDs. (**A**). The synthetic route of AOCDs and AOCDs:Ca, (**B**). AOCDs particle size and (**C**). AOCDs:Ca particle size detection by TEM, The lines mark the adjacent diffraction peaks and indicate them with arrows (**D**). AOCDs and Ca @ AOCDs fluorescence emission spectra, (**E**). AOCDs and AOCDs:Ca wavelength redshift, (**F**). UV absorption spectra, (**G**). Fluorescence lifetime.

**Figure 3 antioxidants-12-00583-f003:**
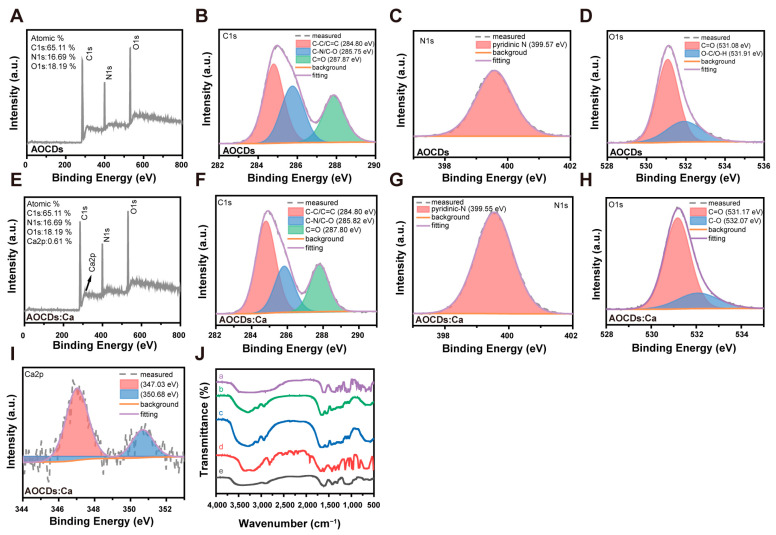
Structural analysis of CDs. XPS analyzed the structure of AOCDs (**A**). Element content; (**B**). C1s spectra; (**C**). N1s spectra; (**D**). O1s spectra, XPS analyzed the structure of AOCDs:Ca; (**E**). Element content; (**F**). C1s spectra; (**G**). N1s spectra; (**H**). O1s spectra; (**I**). Ca2 p spectra; (**J**). Comparison of raw materials and CDs structure by FT-IR. a: Calcium gluconate; b: AOCDs:Ca; c: AOCDs; d:Acryamide; e: Carboxymethyl chitosan.

**Figure 4 antioxidants-12-00583-f004:**
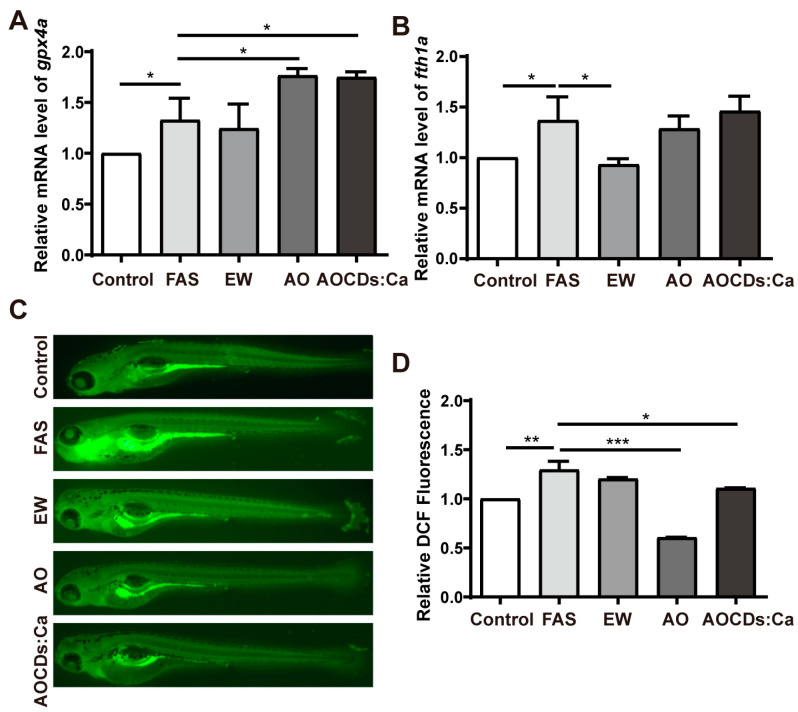
Antioxidant properties of AOCDs and AOCDs:Ca. (**A**). Relative expression level of gpx4a, (**B**). Relative expression level of fth1a, (**C**). Qualitative detection of ROS level, (**D**). Relative quantitative analysis of ROS content (*n* = 3, *t*-test, * *p* < 0.05, ** *p* < 0.01, *** *p* < 0.001).

**Figure 5 antioxidants-12-00583-f005:**
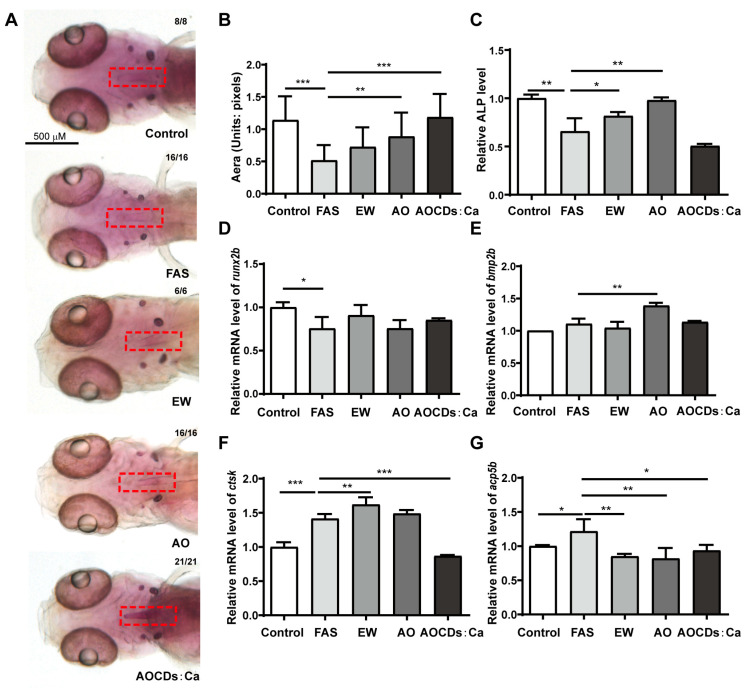
AOCDs:Ca treated iron overload osteoporosis by calcium supplementation. (**A**). Alizarin red staining., The red rectangle is the location of osteoporosis. (**B**). Numerical analysis of staining area (unit: pixel), (**C**). The content of ALP, Effect of treatment on genes osteoblast-regulated genes (**D**). runx2b, (**E**). bmp2b, (**F**). ctsk, (**G**). acp5b, (*t*-test, * *p* < 0.05, ** *p* < 0.01, *** *p* < 0.001).

**Figure 6 antioxidants-12-00583-f006:**
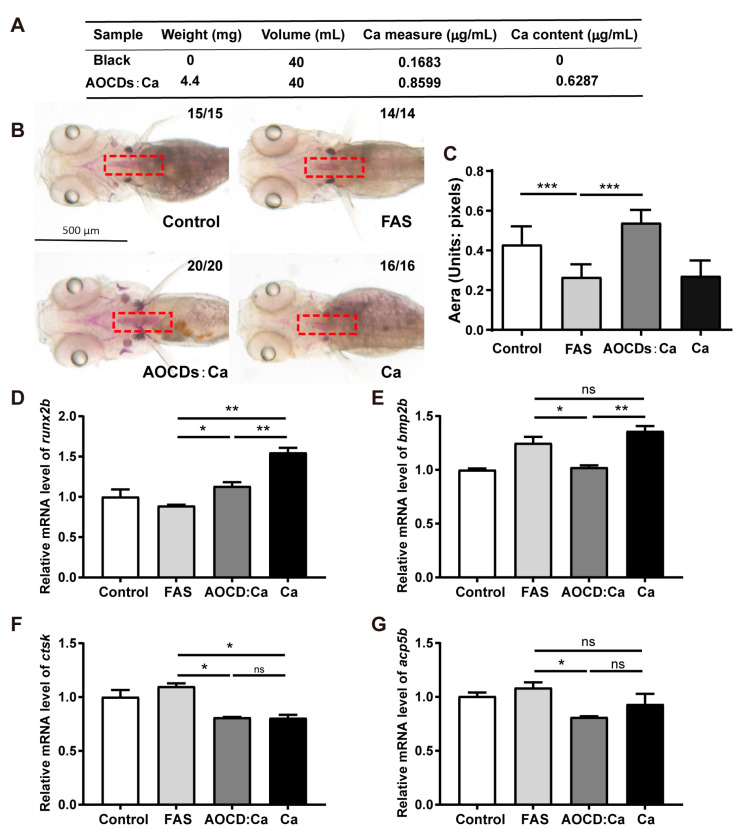
The effect of AOCDs:Ca superior to calcium in iron overload OP. (**A**). The measure of calcium content in AOCDs:Ca, (**B**). Alizarin red staining, (**C**). Numerical analysis of staining area (unit: pixel), (*n* = 3, t-test, *** *p* < 0.001). Effect of treatment on genes osteoblast-regulated genes, (**D**). runx2b, (**E**). bmp2b, (**F**). ctsk, (**G**). acp5b, (*t*-test, * *p* < 0.05, ** *p* < 0.01, *** *p* < 0.001).

## Data Availability

Data is contained within the article and in Appendix A.

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
