# Peer review of "Antioxidant Carboxymethyl Chitosan Carbon Dots with Calcium Doping Achieve Ultra-Low Calcium Concentration for Iron-Induced Osteoporosis Treatment by Effectively Enhancing Calcium Bioavailability in Zebrafish"

_antioxidants, 2023, doi:10.3390/antiox12030583_

Round 1

Reviewer 1 Report

The authors use zebrafish to unravel the potential of novel compounds to treat osteoporosis. The design and results are very interesting and can be of use to the community. However, I have some major concerns about sections:

-       In the introduction the authors claim that “AOCDs:Ca is a potential drug for the treatment of iron overload OP, especially post-menopausal women OP”. I consider zebrafish as an ideal model for these experiments, and the results are compelling. However, I would like to suggest the authors to tone down these statements. There is a huge difference between a 6 day old embryo and a post-menopausal woman. Furthermore, this experimental design does not specifically address osteoporosis but bone ossification. When the authors add their drugs, bones are developing. Please, be clear in the text about this issue.

-       The authors expose the embryos to FAS at 4 days (I assume). At this stage bone formation is starting. Osteoporosis is the loss of bone, typically in elderly people. While I think this is a robust model to test the author´s hypothesis, they should show that the embryos are at least healthy and there is not any gross developmental delay. For example, add pictures of life embryos and you could perform Alcian Blue staining to observe the cartilage that is partially supporting ossification.

-       It is difficult to see any difference in the Alizarin Red staining in Figure 1. Furthermore, FAS-treated embryo is way lighter, what could account for the differences. In fact, in the methods section authors claim to use a Macro zoom fluorescence microscope on samples with no obvious color. Then, I am not sure what I am looking at in figure 1. Please include the fluorescence images and make the quantification with them, light microscopy is not robust enough for these experiments.

-       Also, alizarin red is quantified differently in figures 1 and 5, at least the units in the Y axis are different. Again, in Figure 5 the difference in the alizarin staining cannot be observed. Please include a higher magnification fluorescence image of the bone of interest. It is also weird that the value for alizarin red in figure 6 are very different to figure 5. What is the reason for these differences? Are measurements standardized differently? Is it because the scale difference?

-       According to the methods ROS is measured by some kit in homogenized embryos but in figure 4C we can see whole embryos. What is exactly that ROS? MItosox? How were those embryos stained? Also, I assume MDA means malondialdehyde for lipid peroxidation. Please, clarify

-       Also, please clarify what are EWCDs and how they were used.

I have some other minor comments:

-       The first part of the methods should be deleted since they are instructions

-       “Zebrafish embryos were randomly placed in 6-well culture plates (30 embryos in 4mL solution per well). Deionized water was added to the control group”. I assume embryos where in E3 medium and deionized water was added to the control instead of FAS. It is difficult for the embryos to survive in just water

-       What time was FAS added? Text indicates that solution was changed every 24 hours and embryos were reared for 48 hpf. Since experiments end up at 6 dpf, I assume FAS added at 4 dpf. Alternatively, it was added at 4 hpf and then remove at 2 dpf. Please clarify. Also, AOCDs are added at 3-4 hpf, and I suppose they are maintained until the end of the experiment at 6 dpf. So, when do they exactly overlap? In the graphs, every compound is separate as if they were independent treatments. Please, check and change accordingly if needed. Maybe a small cartoon would help.

-       it is described that ROS is quantified in homogenized embryos and some kit. Please, include the reference of those kits

-       Please include de Ca treatment in the method section

-       Please, do not say “head bone”. Bones have names, identify them properly

-       Please be careful with the nomenclature. Genes should be in italic also in the text.

-       “The frequency of autonomic movement, heart rate, hatching rate and survival rate were no significant difference”. Please include life images of the embryos exposed to AOCDs and AOCDs:Ca together with the quantifications.

-       Statements such as the following one should be supported by references. “It is well known that the metabolic balance of bone is maintained by osteoblasts and osteoclasts. Although EWCDs promoted the production of ALP, it also up-regulated the expression of osteoclast gene. This may be the reason why EWCDs does not treat iron overload OP”. There are several examples along the text.

-       The first two paragraphs of the Conclusion look like a recollection of bullet points. It is just a little bit more than repeating results and how your results compare to others is not clear

Author Response

-       In the introduction the authors claim that “AOCDs:Ca is a potential drug for the treatment of iron overload OP, especially post-menopausal women OP”. I consider zebrafish as an ideal model for these experiments, and the results are compelling. However, I would like to suggest the authors to tone down these statements. There is a huge difference between a 6 day old embryo and a post-menopausal woman. Furthermore, this experimental design does not specifically address osteoporosis but bone ossification. When the authors add their drugs, bones are developing. Please, be clear in the text about this issue.

Response: We agree with the comment. We have revised the introduction to the manuscript. There is a huge difference in osteoporosis between 6 dpf zebrafish and postmenopausal women. The osteoporosis in this study is mainly reflected by the degree of bone ossification becoming. Although, both phenotypes are ossification reduction. However, the zebrafish is in the stage of bone development and his ossification percentage is mainly related to the developmental period. In post-menopausal women osteoporosis is mainly due to calcium loss. We have modified the presentation of AOCDs on the treatment of osteoporosis in postmenopausal women to make the article more rigorous.

-       The authors expose the embryos to FAS at 4 days (I assume). At this stage bone formation is starting. Osteoporosis is the loss of bone, typically in elderly people. While I think this is a robust model to test the author´s hypothesis, they should show that the embryos are at least healthy and there is not any gross developmental delay. For example, add pictures of life embryos and you could perform Alcian Blue staining to observe the cartilage that is partially supporting ossification.

Response: We are grateful for the suggestion. FAS can cause reduced ossification, resulting in osteoporosis. However, whether this is achieved by inhibiting zebrafish embryonic development needs to be ruled out. To this end, we reared zebrafish, divided into Control and FAS groups, and counted autonomous movements at 24h, heartbeat counts at 48h, hatching rates at 72h and mortality rates at 96h (Supplemental Figure 3). The results of the experiment showed that the development of the FAS and Control groups was basically the same, proving that FAS did not have a significant effect on zebrafish development. The results of this experiment were added to the supplementary material and the results are plotted below.

Supplemental Figure 3. Effect of FAS on zebrafish embryonic development. A. 24 hpf exercise frequency, B. 48 hpf heart rate, C. 72 hpf hatching rate, D. 96 hpf survival rate (t-test, n=30, repeated 3 times, *P<0.05, **P<0.01, ***P<0.001).

-       It is difficult to see any difference in the Alizarin Red staining in Figure 1. Furthermore, FAS-treated embryo is way lighter, what could account for the differences. In fact, in the methods section authors claim to use a Macro zoom fluorescence microscope on samples with no obvious color. Then, I am not sure what I am looking at in figure 1. Please include the fluorescence images and make the quantification with them, light microscopy is not robust enough for these experiments.

Response: In fact, we did not take fluorescence photographs of the zebrafish in Alizarin Red staining, just plain optical imaging. We used the Macro zoom fluorescence microscope due to its greater magnification and greater clarity. Alizarin Red staining of zebrafish is actually quite clear and gives a visual indication of the degree of bone mineralisation. It is just not obvious in the Figure 1 as chosen and we have replaced it with another one from the same group.

-       Also, alizarin red is quantified differently in figures 1 and 5, at least the units in the Y axis are different. Again, in Figure 5 the difference in the alizarin staining cannot be observed. Please include a higher magnification fluorescence image of the bone of interest. It is also weird that the value for alizarin red in figure 6 are very different to figure 5. What is the reason for these differences? Are measurements standardized differently? Is it because the scale difference?

Response: We apologize for the different units in the Y axis. We used Image J to perform statistical analysis of the images. But we don't know why the statistics vary so much between batches of images. In fact, this does not affect the comparison between groups of the same batch and the conclusions are fine. For the uniformity of the data, we further normalised the values in Figures 1, 5 and 6 with Control as a reference.

-       According to the methods ROS is measured by some kit in homogenized embryos but in figure 4C we can see whole embryos. What is exactly that ROS? MItosox? How were those embryos stained? Also, I assume MDA means malondialdehyde for lipid peroxidation. Please, clarify

Response: We apologize for the many errors in the method ‘Determination of ROS and ALP’. In fact, the ROS assay was performed on intact zebrafish, while the homogenized embryos used for ALP, and the experiment did not involve the detection of MDA. The rewrite reads as follows.

2.8 Determination of ROS.

Live Zebrafish were added with 0.1mM DCFH-DA, kept in the dark for 1 h, and washed with DW. After zebrafish embryos were anesthetized with 0.05% Tricaine for 5 min, the zebrafish was observed and photographed by fluorescence microscope.

2.9 Determination of ALP.

30 zebrafish embryos were collected and homogenized with 0.9% NaCl. Centrifugation was performed at 1000 g/min at 4℃ for 10 min. The total protein assay kit (BCA method) was purchased by beyotime (Jiangsu, China). The content of ALP were measured using Kit (Nanjing Jian cheng).

-       Also, please clarify what are EWCDs and how they were used.

Response: EWCDs are a type of carbon dot previously synthesised in our laboratory that can treat NAFLD. For more information on the synthesis and therapeutic mechanism of EWCDs you can consult our previously published article [1]. As EWCDs have the same antioxidant effect as AOCDs, they are used in this article as a comparison. EWCDs are used in exactly the same way as AOCDs and we add further information about this in the manuscript.

 [1] Yu, L.D.; He, M.Y.; Liu, S.H.; Dou, X.Y.; Li, L.; Gu, N.; Li, B.S.; Liu, Z.G.; Wang, G.X.; Fan, J.L. Fluorescent Egg White-Based Carbon Dots as a High-Sensitivity Iron Chelator for the Therapy of Nonalcoholic Fatty Liver Disease by Iron Overload in Zebrafish. ACS Appl. Mater. Interfaces 2021, 13, 54677-54689.

I have some other minor comments:

-       The first part of the methods should be deleted since they are instructions

-       “Zebrafish embryos were randomly placed in 6-well culture plates (30 embryos in 4mL solution per well). Deionized water was added to the control group”. I assume embryos where in E3 medium and deionized water was added to the control instead of FAS. It is difficult for the embryos to survive in just water

Response: We do not know if E3 water will interact with the carbon dots and therefore deionised diluted carbon dots are the best choice. Therefore the control group should also be reared in deionised water. Although, in general, people choose E3-reared zebrafish embryos as a control, the fact is that zebrafish embryos can survive for a long time in deionised water.

-       What time was FAS added? Text indicates that solution was changed every 24 hours and embryos were reared for 48 hpf. Since experiments end up at 6 dpf, I assume FAS added at 4 dpf. Alternatively, it was added at 4 hpf and then remove at 2 dpf. Please clarify. Also, AOCDs are added at 3-4 hpf, and I suppose they are maintained until the end of the experiment at 6 dpf. So, when do they exactly overlap? In the graphs, every compound is separate as if they were independent treatments. Please, check and change accordingly if needed. Maybe a small cartoon would help.

Response: We agree with the comment. We have revised the method to the manuscript. The rewrite reads as follows.

Zebrafish embryos were randomly placed in 6-well culture plates (30 embryos in 4mL solution per well). Deionized water was added to the control group. The model group was added 0.2 mM ferrous ammonium sulfate (FAS, Aladdin). The solution was changed every 24 h. Zebrafish embryos in FAS were reared at 28℃ for 48 h. Then embryos were rinsed with deionized water. Deionized water was added to the control group. The model group was added carbon dots solution at 48 hpf without water change. All group zebrafish larvae were raised to 6 dpf.

-       it is described that ROS is quantified in homogenized embryos and some kit. Please, include the reference of those kits. -       Please include de Ca treatment in the method section。-       Please, do not say “head bone”. Bones have names, identify them properly. -       Please be careful with the nomenclature. Genes should be in italic also in the text.

Response: We agree with the comment and make corrections.

-       “The frequency of autonomic movement, heart rate, hatching rate and survival rate were no significant difference”. Please include life images of the embryos exposed to AOCDs and AOCDs:Ca together with the quantifications.

Response: Toxicology-related experiments are data recorded by visual observation under a light microscope. Unless the toxicity causes deformities in the zebrafish, it is meaningless to simply take photographs. If necessary we can recreate a batch of models and videotape them while counting. However, we do not believe that there is much point; the experiment is simple and easy to perform and not technical.

-       Statements such as the following one should be supported by references. “It is well known that the metabolic balance of bone is maintained by osteoblasts and osteoclasts. Although EWCDs promoted the production of ALP, it also up-regulated the expression of osteoclast gene. This may be the reason why EWCDs does not treat iron overload OP”. There are several examples along the text.

Response: We agree with the comment and make corrections.

-       The first two paragraphs of the Conclusion look like a recollection of bullet points. It is just a little bit more than repeating results and how your results compare to others is not clear

Response: We agree with the comment. We have revised the manuscript. The rewrite reads as follows.

FAS was selected to establish the iron overload OP model in zebrafish. Because, FAS can better cause reduced ossification without affecting the development of zebrafish, thus creating a model of osteoporosis (Supplemental Figure 3 and 4). When iron overload is mentioned, the first thing that comes to mind is treatment with iron chelators. However, chelators are often accompanied by side effects [40-43]. Therefore, it is an important research topic to find a safer and more effective iron chelator.

Reviewer 2 Report

The manuscript regarding the topic and results presented is of interest to the scientific community and major revisions based on the comments below are recommended before considering for publication.

1: The title should be abbreviated since it is too lengthy and dispersive and does not give an idea of the work.

2: The introduction should be revised. The authors wrote: "Increased iron levels in postmenopausal women may be a  cause of OP." This issue should be discussed in more detail in the introduction.

3: The authors should recheck the statistical analysis of the figure 1. in particular in figure 1B, the results do not seem to be significant; figure 1E (relative ALP level) does not seem to have the significance shown,  as well as  figure 1G.

4: There is no figure 3 in the work

5: Figures 5 and 6 need to be corrected in bold

6: Figure 6 needs to be moved before conclusions

7: Conclusion and outlook should be changed as discussion and conclusion since the results paragraphs were not discussed.

Author Response

Comments and Suggestions for Authors

The manuscript regarding the topic and results presented is of interest to the scientific community and major revisions based on the comments below are recommended before considering for publication.

Response: We do appreciate it for reviewer’s positive suggestion. We have revised the manuscript totally according to all reviewer’s suggestion and comments.

1: The title should be abbreviated since it is too lengthy and dispersive and does not give an idea of the work.

Response: In our revised version, we have corrected these tediums and tried to keep it short and interesting.

2: The introduction should be revised. The authors wrote: "Increased iron levels in postmenopausal women may be a cause of OP." This issue should be discussed in more detail in the introduction.

Response: Thank you for your close reading of our paper. This is an important point, so we have modified the introduction.

Estrogen inhibits iron-regulator synthesis, maintains iron transporter protein integrity and enhances iron uptake by duodenal enterocytes and iron release from iron storage macrophages and hepatocytes. Postmenopausal women therefore exhibit iron accumulation outside of estrogen deficiency. Iron accumulation promotes bone resorption and bone loss through oxidative stress and inflammatory responses. [13,15-18]

References

  1. Zhang, P.; Wang, S.; Wang, L.; Shan, B.C.; Zhang, H.; Yang, F.; Zhou, Z.Q.; Wang, X.; Yuan, Y.; Xu, Y.J. Hepcidin is an endogenous protective factor for osteoporosis by reducing iron levels. J. Mol. Endocrinol. 2018, 60, 299-308, doi:10.1530/jme-17-0301.
  2. Eastell, R.; O'Neill, T.W.; Hofbauer, L.C.; Langdahl, B.; Reid, I.R.; Gold, D.T.; Cummings, S.R. Postmenopausal osteoporosis. Nature Reviews Disease Primers 2016, 2, 16069, doi:10.1038/nrdp.2016.69.
  3. Tsay, J.; Yang, Z.; Ross, F.P.; Cunningham-Rundles, S.; Lin, H.; Coleman, R.; Mayer-Kuckuk, P.; Doty, S.B.; Grady, R.W.; Giardina, P.J.; Boskey, A.L.; Vogiatzi, M.G. Bone loss caused by iron overload in a murine model: importance of oxidative stress. Blood 2010, 116, 2582-2589, doi:10.1182/blood-2009-12-260083.
  4. Zhang, J.; Zhao, H.; Yao, G.; Qiao, P.; Li, L.; Wu, S. Therapeutic potential of iron chelators on osteoporosis and their cellular mechanisms. Biomed. Pharmacother. 2021, 137, 111380, doi:10.1016/j.biopha.2021.111380.
  5. Zhang, H.; Wang, A.; Shen, G.; Wang, X.; Liu, G.; Yang, F.; Chen, B.; Wang, M.; Xu, Y. Hepcidin-induced reduction in iron content and PGC-1β expression negatively regulates osteoclast differentiation to play a protective role in postmenopausal osteoporosis. Aging 2021, 13, 11296-11314, doi:10.18632/aging.202817.

3: The authors should recheck the statistical analysis of the figure 1. in particular in figure 1B, the results do not seem to be significant; figure 1E (relative ALP level) does not seem to have the significance shown, as well as figure 1G.

Response: Thank you for this observation. We used Image J to perform statistical analysis of the images. But we don't know why the statistics vary so much between batches of images. In fact, this does not affect the comparison between groups of the same batch and the conclusions are fine. For the uniformity of the data, we further normalised the values in Figures 1with Control as a reference.

4: There is no figure 3 in the work

Response: In our revised version, we corrected this error. In the revised manuscript, we rephrased the corresponding sentences to avoid misleading.

Figure 3. Structural analysis of CDs. XPS analyzed the structure of AOCDs A. Element content B. C1s spectra, C. N1s spectra, D. O1s spectra, XPS analyzed the structure of AOCDs:Ca, E. Element content, F. C1s spectra, G. N1s spectra, H. O1s spectra, I. Ca2 p spectra, J. Comparison of raw materials and CDs structure by FT-IR.

In order to further study AOCDs and AOCDs:Ca functional groups on the surface, we analyzed the AOCDs and AOCDs:Ca chemical structure by X-ray photoelectron spectroscopy (XPS) and Fourier transform infrared (FT-IR) spectra. The full XPS spectra of AOCDs showed three peaks of 284.57, 398.65 and 532.85 eV, indicating that AOCDs was composed of C, O and N elements with atomic ratios of 65.11%, 18.19% and 16.69%, respectively (Figure 3A). The high-resolution XPS spectra of the C 1s band were divided into three peaks at 284.80, 285.75 and 287.87 eV, corresponding to C-C/C=C, C-N/C-O and C=O, respectively (Figure 3B). N 1s band had one peak located at 399.57 eV, which belonged to Pyridine N (Figure 3C) [26]. The O 1s band had two peaks at 531.08 and 531.91 eV, corresponding to C=O and O-C/O-H, respectively (Figure 3D). The total XPS spectra of AOCDs:Ca showed 284.57, 398.65, 532.85 and 346.78 Ca2p eV peaks, indicating that AOCDs:Ca was composed of C, O, N and Ca elements, with atomic ratios of 49.09%, 33.23%, 17.07% and 0.61%, respectively (Figure 3E). The high-resolution XPS energy spectra of C 1s band was divided into three peaks at 284.80, 285.82 and 287.80 eV, corresponding to C-C/C=C, C-N/C-O and C=O, respectively (Figure 3F) . N 1s band had one peak located at 399.55 eV, which belonged to Pyridine N (Figure 3G) . The O 1s band had two peaks at 531.17 and 532.07 eV, corresponding to C=O and C-O, respectively (Figure 3H) . The Ca 2p band had two peaks located at 347.03 and 350.68 eV (Figure 3I). Together, these four high-resolution spectra showed that Ca atoms had been successfully embedded in AOCDs:Ca and existed as ions. This provided the basis for the efficient release of Ca2+. In addition, FT-IR spectra showed that the stretching vibration of O-H/N-H from CC of 3291 cm-1 and the stretching vibration of carboxyl (1650 and 1438 cm-1) and amide carbonyl from AM (C=O, 1650 cm-1) in AOCDs. The mixed in-plane bending vibration of amides C-H and N-H was located at 1436 cm-1. Therefore, AOCDs and AOCDs:Ca are obviously hydrophilic. Based on the above characterization results, Ca2+ doped success, AOCDs:Ca have aromatic structure similar to AOCDs. It also has the optical properties of the origin of the defect of pyridine nitrogen atom (Figure 3J).

5: Figures 5 and 6 need to be corrected in bold

Response: In our revised version, we corrected this error. In the revised manuscript, we rephrased the corresponding sentences to avoid misleading.

6: Figure 6 needs to be moved before conclusions

Response: We are very grateful to the reviewer’s careful check. In the revised manuscript, we rephrased the corresponding sentences to avoid misleading.

7: Conclusion and outlook should be changed as discussion and conclusion since the results paragraphs were not discussed.

Response: We thank the reviewer for the careful check and nice suggestions. In the revised manuscript, we rephrased the corresponding sentences to avoid misleading.

Round 2

Reviewer 1 Report

Thanks for the response. These compounds will be useful for other researchers.

In relation to the differences in the quantification of the alizarin red, I would suggest, for the future, that authors use fluorescence. In my hands light microscopy might be quite variable in different days. Also Fiji has some plugins that help with these differences.

Reviewer 2 Report

In my opinion the manuscript can be considered for publication